# Association of Infant Feeding Patterns with Taste Preferences in European Children and Adolescents: A Retrospective Latent Profile Analysis

**DOI:** 10.3390/nu11051040

**Published:** 2019-05-09

**Authors:** Elida Sina, Christoph Buck, Hannah Jilani, Michael Tornaritis, Toomas Veidebaum, Paola Russo, Luis A. Moreno, Denes Molnar, Gabriele Eiben, Staffan Marild, Valeria Pala, Wolfgang Ahrens, Antje Hebestreit

**Affiliations:** 1Leibniz Institute for Prevention Research and Epidemiology—BIPS, Achterstr. 30, 28359 Bremen, Germany; sina@leibniz-bips.de (E.S.); buck@leibniz-bips.de (C.B.); jilani@leibniz-bips.de (H.J.); ahrens@leibniz-bips.de (W.A.); 2Institute for Public Health and Nursing Research—IPP, University of Bremen, 28359 Bremen, Germany; 3Research and Education Institute of Child Health, 2035 Lefcosia, Cyprus; tor.michael@cytanet.com.cy; 4Department of Chronic Diseases, National Institute for Health Development, 11619 Tallin, Estonia; toomas.veidebaum@tai.ee; 5Institute of Food Sciences, National Research Council, 83100 Avellino, Italy; prusso@isa.cnr.it; 6GENUD (Growth, Exercise, Nutrition and Development) Research Group, Instituto Agroalimentario de Aragón (IA2), Instituto de Investigación Sanitaria Aragón (IIS Aragón), Centro de Investigación Biomédica en Red Fisiopatología de la Obesidad y Nutrición (CIBERObn), University of Zaragoza, 50009 Zaragoza, Spain; lmoreno@unizar.es; 7Department of Pediatrics, Medical School, University of Pécs, 7623 Pécs, Hungary; denes.molnar@aok.pte.hu; 8Department of Biomedicine and Public Health, School of Health and Education, University of Skövde, 54128 Skövde, Sweden; gabriele.eiben@his.se; 9Department. of Pediatrics, Institute of Clinical Sciences, Sahlgrenska Academy at University of Gothenburg, 40530 Gothenburg, Sweden; staffan.marild@pediat.gu.se; 10Department of Preventive and Predictive Medicine, Fondazione IRCCS, Istituto Nazionale dei Tumori, 20133 Milan, Italy; Valeria.Pala@istitutotumori.mi.it; 11Faculty of Mathematics/Computer Science, University of Bremen, 28359 Bremen, Germany

**Keywords:** breastfeeding, formula milk, taste preference, healthy diet adherence, children, IDEFICS study, I.Family

## Abstract

The aim was to investigate associations between the duration of infant feeding practices (FP) and taste preferences (TP) in European children and adolescents. A total of 5526 children (6–16 years old) of the I.Family study completed a Food and Beverage Preference Questionnaire to measure their preferences for sweet, fatty and bitter tastes. Mothers retrospectively reported the FPs duration in months: exclusive breastfeeding (EBF), exclusive formula milk feeding (EFMF), combined breastfeeding (BF&FMF) and the age at the introduction of complementary foods (CF). Using logistic regression analyses and latent class analysis (latent profiles of FP and CF were identified), we explored associations between profiles and TP, adjusting for various covariates, including the Healthy Diet Adherence Score (HDAS). A total of 48% of children had short durations of EBF (≤4 months) and BF&FMF (≤6 months) and were introduced to CF early (<6 months). No significant relationship was observed between the single FPs and TP, even when considering common profiles of FP. HDAS was inversely associated with sweet and fatty TP, but positively with bitter TP. Contrary to our hypotheses, we did not observe associations between FP and children’s TP later in life. Further studies with higher FP variation and longitudinal design are needed to investigate the causal associations between infant FP and taste preferences later in life.

## 1. Introduction

Taste preference (TP) is one of the factors that affect the children’s food intake and eating habits [1]. Humans can perceive 6 main basic tastes: (1) Sweet taste is caused by sugar and its derivatives such as fructose or lactose, but other substances such as amino acids and alcohol in fruit juices or alcoholic drinks can also activate the sensory cells that respond to sweetness; (2) Sour taste is mostly perceived via acidic solutions such as lemon juice or organic acids and is caused by hydrogen ions; (3) Salty taste is mainly perceived through foods containing table salt. Its chemical basis is the salt crystal, which consists of sodium and chloride. The sensation of saltiness can be caused by other mineral salts such as potassium or magnesium salts [2]; (4) Bitter taste is brought by a variety of components such as 6-*n*-propylthiouracil (PROP), sinigrin and goitrin, found in cruciferous vegetables (e.g., broccoli). There are 25 bitter taste receptors in humans but the most studied is *TAS2R38*. Genetic variations in this receptor cause different responses in taste sensitivity to bitter compounds from one human to another [3]; (5) Umami taste is caused by glutamic acid or aspartic acid and is similar to the taste of meat broth. It is also found in some plants, such as ripe tomatoes or asparagus [4]; (6) Fatty taste, called Oleogustus, has been described as the sixth basic taste. The stimuli devoted to the detection of dietary fat taste are the Non-Esterified Fatty Acids (NEFA). In particular, medium and long-chain fatty acids have a distinct taste sensation compared to other basic tastes (sweet, bitter, sour and salty) [5,6]. 

Evidence for the influencing role of genetic and environmental factors on the development of TP is well established [7]. Infants prefer the sweet taste and reject the sour and bitter tastes [7], while the preference for salt appears at about 4 months postnatally [8,9]. TP are learned during contact with food and the eating environment. An infant’s experience with flavors begins in the mother’s womb and during lactation, when flavors from the mother’s diet are transmitted to her amniotic fluid, and later to her colostrum and milk. The infant-feeding method parents choose, whether it is breast or formula milk, will later influence their child-feeding practices [10,11] and the development of their children’s food preferences and food acceptance patterns [12]. 

The fundamental role of breastfeeding on different physiological functions and on the infant’s early immunity has been recognized through international nutritional policies, such as the World Health Organization guidelines on early life feeding, which recommends the exclusive breastfeeding of infants up to the age of 6 months and at least for the first 4 months of life [13]. Exclusive breastfeeding is crucial for the growth and development of infants [14], has a long term impact in shaping children’s eating behaviors, and predicts the Body Mass Index (BMI) during childhood [15] and later in adult life [16]. Previous studies have pointed out the positive influence of breastfeeding duration on food variety and higher intake of fruits and vegetables in preschoolers [17,18], including in 4 European cohorts [19] and in school-aged children [20,21]. Vital compounds in the human milk provide a specific taste, such as lactose for the sweet taste, glutamate for the umami or savory taste, sodium for the salty taste, urea for the bitter taste and long-chain fatty acids for the fatty taste [22,23].

The introduction of formula milk and other complementary foods represents a crucial period for establishing infants’ taste preferences and attitudes towards food, as well as for obesity prevention [24]. Children who were fed exclusively with formula milk do not benefit from the rich flavor profile of their mother’s milk: their flavor experience is poorer as they don’t experience the flavors from the variety of foods in the mother’s diet. Various types and brands of formula milk products offer a diversity of flavors: milk-based formulas are described as having low levels of sweetness and ‘sour and cereal-type’; soy-based formulas are described as tasting sweeter, more sour and bitter, and having a ‘hay/beany’ odor, whereas the hydrolysate formulas are extremely unpalatable to adults due to their sourness and bitterness [25]. Formula-fed infants learn to prefer the flavors associated with the formula milk they were fed and this has been found to influence taste preferences later in life [3,25]. Infant formulas might differ in protein, fat and carbohydrate composition and/or structure, and these differences may, in turn, affect growth, health outcomes and taste preferences [26]. 

Scott et al. demonstrated that breastfeeding duration is directly associated with the food variety at two years of age [18], independent of factors that are known to influence diet quality in children, such as maternal age and education [27,28]. Another study found that having been breastfed was positively associated with a healthier dietary pattern amongst older Australian children [20]. Burnier et al. [29] investigated longitudinal data from the Quebec Longitudinal Study of Child Development and observed that 3 or more months of exclusive breastfeeding appeared to be a predictive factor for the higher consumption of vegetables in preschool age children. Nicklaus and colleagues found that 2–8 year old children who were breastfed for at least three or more months were more likely to eat vegetables compared to those who were breastfed for a shorter time [30]. A number of animal studies [31,32] and experimental studies in humans [33,34] indicated that breastfeeding is associated with a greater acceptability of new food and flavors during the weaning period. Breastfeeding can contribute towards reducing infants’ fears to try new foods and facilitate the transition from milk feeding to solid food eating with lower resistance. Consequently, this can lead to an intake of a higher food variety in breastfed children [3,21,35,36]. 

Although research has shown that breastfeeding influences infants’ food acceptance [22,37,38,39], to our knowledge, no studies have examined whether it shapes taste preferences in later stages of life. This study seeks to fill this gap by examining how breastfeeding practices—in comparison to formula milk feeding—during infancy, affect food TP in later childhood and adolescence, in a population-based cohort of normal, healthy developing children aged 6 to 16 years old, in 7 European countries: Cyprus, Estonia, Germany, Hungary, Italy, Spain and Sweden.

To be exact, we examined the association between infant feeding practices duration (FP): (1) exclusive breastfeeding (EBF), (2) exclusive formula milk feeding (EFMF) and (3) combined strategy (BF&FMF), and taste preferences (TP) evaluated in our study: sweet, fatty and bitter. We further considered a latent class analysis to identify feeding patterns from a combination of feeding practices and food introduction, and their association with taste preferences.

## 2. Materials and Methods 

### 2.1. Study Sample

I.Family builds on the IDEFICS (Identification and prevention of Dietary and lifestyle-induced health Effects in Children and InfantS) study, whereby it additionally engages the families of the children who were examined during the IDEFICS baseline (T0) and/or follow-up survey (T1) [40,41]. During the IDEFICS study in 2007/2008, 16,229 children aged 2–9 years from Belgium, Cyprus, Estonia, Germany, Hungary, Italy, Spain and Sweden participated in the baseline survey (T0). Two years later (T1), 13,596 children were examined, 11,041 of whom had previously participated in T0 (68%). In 2013/2014, I.Family (T3) collected further data on the lifestyle-related diseases of 7105 (52% of T1) children, who were then between 7 and 17 years old [42]. Data of 5526 children aged 6–16 years old who completed the Food and Beverage Preference Questionnaire (FBP) during I.Family were used. For the purpose of this investigation, Belgian participants were excluded as their data on food and beverage preference were not collected. Further, retrospective information from the Pregnancy and Early Childhood Questionnaire concerning breastfeeding and formula milk feeding practices and their respective durations during infancy and early childhood reported by mothers in both the IDEFICS study and I.Family were linked. Parents provided written informed consent for all examinations. Each child was informed orally about the measurements by field workers and asked for his/her consent immediately before the examination.

### 2.2. Core Questionnaire 

Information on sex, age, country, migration and socio-economic status (SES) of I.Family participants were collected using a self-administered questionnaire. A validated [43,44] and reproducibility tested [45] food frequency questionnaire (FFQ) containing 59 food and beverage items was completed for each participant. The response categories were ‘never/less than once a week’, ‘1–3 times a week’, ‘4–6 times a week’, ‘1 time/day’, ‘2 times a day’, ‘3 times a day’ and ‘I have no idea’. Based on the FFQ data, a Healthy Diet Adherence Score (HDAS) was developed for all 7 countries [46,47], as a proxy-indicator of children’s adherence to healthy dietary guidelines including a high consumption of fruits and vegetables, of whole meals, fish consumption of 2–3 times per week and a reduced intake of refined sugars and fat. The HDAS was used for the present analyses as a continuous variable and ranged from 0 to 50. A higher score represented a higher adherence to healthy dietary guidelines. The self-reported educational level of parents was assessed based on the International Standard Classification of Education (ISCED) [48] and used as a proxy indicator for SES. For the present analyses, it was classified into two main categories: “low-medium education”, and “high education”. Children’s migration background was assessed based on whether the parents were born outside the respective country of residence and recorded as the migration status in the categories “both parents”, “one parent”, “none of the parents”. 

All questionnaires were developed in English and translated into the respective national languages. They were then back-translated into English to check for translation errors. Children aged 12 years and above self-completed the questionnaire, while parents proxy-reported the relevant questions for children below 12 years of age. The cut-off of 12 years was chosen because children have been shown to be reliable reporters of their food intake at this age [49]. 

### 2.3. Food and Beverage Preference Questionnaire

The questionnaire was constructed as part of the I.Family survey to assess the preferences for sweet, fatty, bitter and salty tasting foods and beverages and was administered to children and adolescents. It contained food photographs of 63 items, including single foods (e.g., broccoli, banana, lettuce), mixed foods (e.g., lasagna, donut), condiments (e.g., nougat spread, butter), and drinks (e.g., fruit juice, lemonade). Using a 1–5 point Likert (smiley) scale, children indicated how much they like the taste of the food given in the photograph, with 1 meaning “do not like at all” and 5 meaning “I like it very much”. Participants were also given the chance to indicate if they had never tried (don’t know) a specific food/drink item (Figure 1). A pre-test was conducted in every center to ensure the availability of all food items across countries. Furthermore, the FBP questionnaire has been shown to provide valid data that are useful for characterizing taste phenotypes in epidemiological studies [50]. For the present analyses, the preferences for sweet and fatty taste were used as proxy-indicators for unhealthy foods [51,52], while bitter taste was used as a proxy-indicator for healthy foods [3]. Our study only focused on taste preferences that are linked to current obesogenic dietary intake in children, characterized by nutrient-dense foods high in fat and sugar and low in fiber [53,54], hence, salty and sour taste were not assessed. Umami taste is under discussion for healthy (e.g., tomatoes) and unhealthy food preferences (e.g., crisps); thus we also did not consider the preference for umami taste. For the sake of clarity, the term “taste preference” will be used instead of preferences for sweet, fatty and bitter tasting foods and beverages, hereafter. 

### 2.4. Taste Preference Scores

The food and beverages included in our analyses were chosen based on factor analyses conducted by Jilani and colleagues [55] that assigned foods to respective taste modalities. In accordance with the factor analyses, we computed scores for the liking of three specific taste modalities: sweet, fatty and bitter, by calculating the mean liking of the foods and drinks included in each of the 3 categories. To control for differences of age and sex in the liking of each taste, the scores were first calculated separately for boys under 12 years old, girls under 12 years, boys aged 12 years and above and girls aged 12 years and above. The cut off of 12 years was chosen as the median age where children enter the age of puberty and further physiological and anatomical developments occur [56]. The scores were then merged into one unique score for each taste modality, in order to assess the association between taste preference scores and infant FP, independent of the age and sex of subjects. The sum of the ratings for the foods and drinks was then calculated and divided by the total number of foods and drinks that were included in the specific taste modality group. The taste preference scores were then categorized as “high” vs. “low”, depending on the children’s answers. Based on the median value, those who reported 4 or 5 on the smiley scale were included in the “high” preference category, while children who reported 3 or below on the Likert scale were categorized in the “low” preference category. Due to missing values in the taste preference scores, the sample size varied. For instance, when analyzing the sweet score, the sample size was higher than for the bitter score, because children recognized the sweet tasting foods more than the bitter tasting ones. 

### 2.5. Pregnancy and Early Childhood Questionnaire 

During the IDEFICS surveys (2007/2008 and 2009/2010) and the I.Family survey (2013/2014), mothers were asked to retrospectively report on the feeding strategies they had chosen during infancy and the early life stage of their children. The FP included the duration of
Exclusive breastfeeding (EBF): calculated as the difference in months between age at start of other forms of feeding (formula or other complementary foods) and the age at the start of EBF (at birth); EBF was then classified in categories “None”, “Up to 4 months” and “More than 4 months”,Combined breastfeeding (BF&FMF): calculated as the total duration of breastfeeding after birth (breastfeeding combined with any other type of feeding, formula or complementary foods) and classified as “None“, ”Up to 6 months” and “More than 6 months”.Exclusive formula milk feeding (EFMF): calculated as the difference of total duration of formula milk feeding in combination with other types of feeding and the duration of either EBF or BF&FMF. It was then categorized as “None”, “Up to 6 months”, “More than 6 months”.

The categories for the infant feeding variables were chosen to try and accommodate the restricted sample size for smaller categories and to facilitate the interpretability of the results. The cut off of 4 months for EBF was chosen based on the WHO recommendations to breastfeed infants exclusively with breastmilk for at least the first 4 months of life [13]., As the introduction of complementary foods and the duration of FP in many countries varies, the cut off of 6 months was chosen for the BF&FMF and EFMF. This is in accordance with WHO observations and findings of other studies [13,22]. Mothers also provided information on the age at first introduction of any of the five food categories: cereals (or foods containing rye, wheat or barley), vegetables, fruits, meat and cow milk. These categories were then merged into one unique variable—the minimum age at the first introduction of any food category—which was then categorized as “up to 6 months”, “later than 6 months” and “missing”. This was again based on WHO observations of the introduction of complementary feeding [13]. From here on, we will refer to these categories as complementary food introduction (CF). 

### 2.6. Statistical Analyses 

Descriptive analyses of FP during infancy were conducted by calculating the mean, median and range of the duration of EBF, BF&FMF and EFMF. Furthermore, the following study characteristics were described, i.e., N and proportions, based on categories of BF&FMF for categorical covariates included in the analyses, such as age groups, sex, SES, migration status, the age of the first introduction of complementary food categories, preference scores for each taste modality and country. In order to evaluate the association between each of the three different FP (independent variables) and taste preferences (dependent variables), i.e., sweet, fatty and bitter, logistic regression analyses to calculate Odds Ratios (OR) and 95% confidence limits adjusted for covariates (country, age, sex, HDAS, CF introduction, parents’ migration status and SES) were conducted. As a combined analysis showed a high multicollinearity of feeding practices and did not yield interpretable results, a latent class analysis (LCA) was conducted in order to identify latent profiles considering categories of FP and age of complementary food introduction [57]. LCA was conducted considering three, four or five latent profiles of seven variables (three FP variables and four separate variables for the introduction of food items). The different LCA were compared by considering the Bayesian Information Criterium (BIC) and a clear distinction of latent profiles in terms of conditional probabilities. The chosen profiles were then used in logistic regression models as independent variables for each of the TP to again calculate Odds Ratios and 95% confidence limits adjusted for the remaining covariates. All statistical analyses were performed using the statistical software SAS, version 9.3 (Statistical Analyses System, SAS Institute Inc., Cary, NC, USA). The latent class analysis was conducted using the PROC LCA Macro (version 1.3.2, University Park: The Methodology Center, Penn State, PA, USA) [58] in SAS 9.3. Level of significance was set to α = 0.05.

## 3. Results

### 3.1. Study Characteristics 

The descriptive analyses of the three main feeding practices, as shown in Table 1, indicated that the median for EBF was 4 months, ranging from not breastfed at all to 36 months of EBF. The BF&FMF had a median of 6 months, varying from 0 to 36 months of total breastfeeding (BF combined with other types of feeding). The EFMF had a maximum of 48 months, with a median of 0 months of formula milk feeding. 

Half of the study population was female (Table 2) and the mean age was 11.6 years (SD = 1.9), whereby 53.3 % of the participants were less than 12 years old. Both parents of a small proportion of the children were migrants (5.4%), while 9.8 % had one parent who was a migrant. Half of the children (50.5%) came from highly educated families and 78.1% of them were introduced to complementary foods early (≤6 months). The HDAS ranged from 0 to 44, with a median of 18. 

Almost 42% of the study population was breastfed for up to 6 months, while 43.6% had a longer duration of more than 6 months and 14.4% were never breastfed. Looking at the proportions of taste preference categories (low vs. high) according to BF&FMF, the children reported high sweet and fatty taste preferences, independent of the duration of BF&FMF during infancy. The contrary was however shown for bitter taste preferences. 

### 3.2. Association between Exclusive Breastfeeding and Taste Preference 

The results of the logistic regression analyses showed no associations between EBF and preferences for sweet, fatty and bitter tastes (Table 3). An increase in the HDAS was observed to significantly decrease the chance for high sweet taste preferences (OR = 0.88, 95%CI [0.82; 0.96]) and high-fat taste preferences (OR = 0.88, 95%CI [0.81; 0.95]), while on the other hand significantly increasing the chance for a high bitter taste preference (OR = 1.31, 95%CI [1.20; 1.43]). A late introduction to complementary foods (≤6 months) was found to significantly decrease the odds for a high-fat taste preference (OR = 0.81, 95%CI [0.68; 0.95]). Compared to having parents with a high SES, having parents with a low/medium SES significantly increased the odds for a high-fat taste preference (OR = 1.14, 95%CI [1.009; 1.30]).

### 3.3. Association between Combined Breastfeeding and Taste Preference 

The logistic regression analyses did not show an association between the duration of BF&FMF and taste preferences (sweet, fatty and bitter) (Table 4). As observed for EBF, significant associations were observed between HDAS and taste preferences, with having a higher HDAS significantly decreasing the odds for high sweet and fatty taste preference but significantly increasing the odds for high bitter taste preference (Table 4). A late introduction of complementary foods significantly decreased the odds for a high-fat taste preference (OR = 0.82, 95%CI [0.70; 0.97]) compared to an introduction of complementary foods before the age of 6 months. Having parents with a low/medium SES again significantly increased the odds for a high-fat taste preference (OR = 1.14, 95%CI [1.004; −1.29]). 

### 3.4. Association between Exclusive Formula Feeding and Taste Preference

No significant association was observed between the EFMF duration and preferences for sweet, fatty and bitter taste (Table 5). HDAS was again significantly associated with sweet and fatty taste preferences, (OR = 0.88, 95%CI [0.81; 0.96]) and OR = 0.88, 95%CI [0.81; 0.95]) respectively, as well as with a high bitter taste preference (OR = 1.31, 95%CI [1.20; 1.42]). Furthermore, a late introduction of complementary foods decreased the chance for a high preference of fatty taste (OR = 0.82, 95%CI [0,70; 0.97]) at a later age, compared to an early introduction. Compared to having a high SES background, having a low/medium SES background increased the odds to prefer a high-fat taste (OR = 1.14, 95%CI [1.00; 1.29]). 

### 3.5. Association between Latent Profiles of Feeding Practices and Taste Preferences

With regard to the four latent profiles, the LCA showed the lowest BIC and a clear interpretable distinction of conditional probabilities for the respective variables. Results of the latent class profiles are presented in Table 6, where names of the profiles were chosen according to the highest conditional probabilities. Almost half of the children and adolescents (48%) had a short duration of EBF (up to 4 months), then were breastfed in combination with formula milk and introduced to complementary foods early (before 6 months). About a quarter (24%) were exclusively breastfed for a long period, were never exclusively fed formula milk and had a late introduction to complementary foods (later than 6 months). A total of 14% were fed formula milk as the main alternative to breastmilk and were as well introduced to complementary foods at a later age. Only 13% were exclusively fed formula milk and were introduced to complementary foods early. 

Profile number 4 was considered to reflect the least advisable feeding strategy (no breastfeeding, only formula milk, early introduction) and served as a reference for the logistic regression analyses presented in Table 7. No significant associations of profiles of feeding practices with taste preferences for sweet, fatty and bitter tastes were observed. Again, the HDAS was significantly negatively associated with sweet (OR = 0.88, 95%CI [0.82; 0.96]) and fatty taste preferences (OR = 0.88, 95%CI [0.81; 0.95]) and significantly positively associated with a bitter taste preference (OR = 1.31, 95%CI [1.20; 1.43]). In addition, children from low/medium SES families were observed to have a higher chance of having a high-fat taste preference (OR = 1.14, 95%CI [1.00; 1.29]). 

## 4. Discussion

To our knowledge, this is the first study assessing the association between different infant FPs and children’s taste preferences in later stages in life which included retrospective and current data from 7 European countries. Our results indicate that European children were predominantly breastfed exclusively for at least 4 months, which is in line with the WHO guidelines [13]. In addition, almost half of the subjects had a long duration of BF&FMF and only a minority (13%) was exclusively fed FM. The feeding strategy parents used seemed not to play a role in the development of taste preferences later in life, irrespective of whether it was EBF, EFMF or a combination. Using both single logistic regression analyses and LCA methods, in which latent profiles of all FP and introduction of complementary food categories were identified, a higher quality diet (HDAS) was observed to be associated with lower chances for a high sweet and high-fat taste preference and increased chances for high bitter taste preference. This suggests that current food choices can actually mold children’s preferences for sweet, fat and bitter tastes, independent of their infant feeding patterns. SES also seemed to play a role, as children who came from a lower SES background were more likely to prefer the fatty taste compared to those from a higher SES background. Our results indicate that a variation in food choice and parental education can affect children’s behaviors towards healthy food choices and preferences.

Our findings are supported by the current evidence, which suggests that children have innate preferences for sweet taste as signalers of high energy foods [3,8]. Further, Schwartz et al. reported that infants’ sweet acceptance was not related to longer durations of EBF [22]. In a longitudinal study, Desor and colleagues measured the sweet preference in children at the age of 11–15 years and again when they were 19–25 years of age, and found that the preferred levels of sucrose decreased over time [59]. Other studies have suggested that children learn to prefer flavors associated with a high dietary fat content [60,61]. Previous findings from the IDEFICS study indicated that children from low educational backgrounds tended to eat more high energy-dense foods, such as sugar-rich and fat-rich foods, compared to those whose parents had a high education [62]. In contrast, children of parents with a higher education tended to eat more fruits and vegetables, generally eating less unhealthy foods. They were also more likely to eat breakfast on a daily basis, emphasizing the influence of parental education on children’s eating habits [47,63,64,65,66,67]. Furthermore, a higher number of fruits and vegetables at 14 months has been shown to increase the preference for these foods and improve the quality of the diet at 3.7 years of age [68]. Our results, which are supported by the current evidence, suggest that taste preferences in children are learned via food experience and are significantly influenced by the food choice and diet literacy of parents. Thus, public health education programs should emphasize the role of food variety in shaping children’s preferences for bitter tasting foods in the long term. Particular attention needs to be paid to parents and other caretakers of low to medium socio-economic status families in order to help them reduce their children’s preferences for high energy-dense foods. 

## 5. Strengths and Limitations

One of the main strengths of our study is the large sample size of 5526 children and adolescents from 7 European countries, which allowed us to have a detailed picture of FP and its potential association with taste preferences in later stages of life. The standardized protocol and the pre-test conducted in a subsample of children showed that the Food and Beverage Preference Questionnaire is a feasible instrument for assessing preferences of food and beverages in children and adolescents. 

Furthermore, having information on covariates such as country of residence, age, sex, HDAS, the timing of the first introduction of complementary feeding, parental education level and migration status allowed us to make adjustments and to control for confounding. 

Nevertheless, there are important methodological limits concerning our research. The scale of the taste preference was slightly limited as it was calculated based on measuring the food preference with only 5 points. This limited our ability to clearly distinguish between extreme taste preferences. Further, in the BF&FMF category, information on the proportion of actual formula milk and breastmilk feeding was not provided. Thus, we have to acknowledge this as a limitation as it hampers a critical discussion on the potential effects of a mixed strategy of feeding on taste preference. 

Since mothers self-reported the details of their infants’ FP (age at starting and termination of infant feeding, the timing of the first introduction of complementary foods) and as adolescents tend to self-report a lower preference for energy-dense (fatty and sweet) foods and beverages [69,70], we cannot entirely exclude social desirability bias. As we used retrospective information on feeding practices, recall bias also potentially affected our data. The reproducibility testing of the early infant parameters showed a weak reproducibility of maternal reports on early infant nutrition, a further potential limitation [71]. Nevertheless, research has shown that mothers recall breastfeeding duration accurately [72,73], while the recall of age at introduction of complementary food is less satisfactory [73]. Moreover, the parents of half of our sample had a high educational status, a fact which might also bias the results with regard to socio-economic status. In addition, we did not have information on other confounding factors such as the role of the maternal diet during breastfeeding and the family diet, factors that have been found to influence taste preference and food intake in children [3,55,74]. Lastly, our research was conducted using only cross-sectional data enriched with retrospective information. We strongly recommend further longitudinal research, e.g., through birth cohorts, that can evaluate the effects of FP during infancy and changes of taste preferences during different stages of life, particularly accounting for the interplay between food choice and socio-economic background.

## 6. Conclusions 

In contrast to our hypotheses, we did not observe an association between infant feeding practices and taste preferences in school children and adolescents, neither regarding the single FPs or as a mixed strategy, nor considering common profiles of FP. Instead, the current diet quality through food choice and educational status of parents consistently showed an association with the current taste preference for sweet, fatty or bitter tastes. Hence, further studies with higher FP variations and using longitudinal data are necessary in order to investigate causal associations between infant feeding practices and taste preferences in later stages of life.

## Figures and Tables

**Figure 1 nutrients-11-01040-f001:**
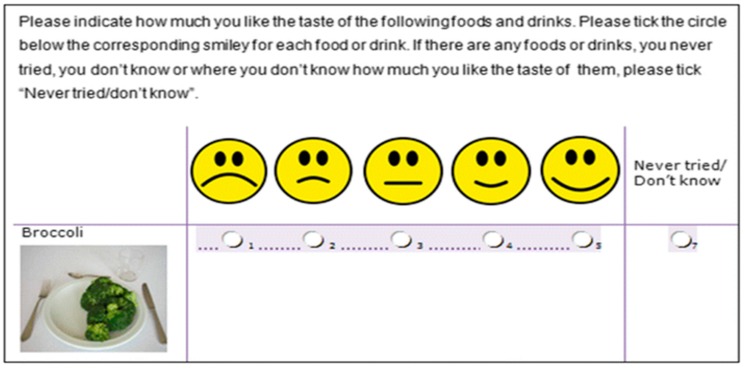
A screenshot of a food item from the Food and Beverage Preference Questionnaire [55].

**Table 1 nutrients-11-01040-t001:** The duration of feeding practices during infancy.

Types of Feeding Practices (N = 5526) Duration in Months	Mean/SD	Median	Min/Max
Exclusive Breastfeeding (EBF)	3.3/2.7	4.0	0.0/36.0
Combined Breastfeeding (BF&FMF)	7.2/6.3	6.0	0.0/36.0
Exclusive Formula Milk Feeding (EFMF)	4.1/8.1	0	0.0/48.0

**Table 2 nutrients-11-01040-t002:** The study characteristics of participants according to the duration of combined breastfeeding (BF&FMF).

	Combined Breastfeeding (BF&FMF)						All	
Variables	None		Up to 6 Months		More Than 6 Months			
	*N*	%	*N*	%	*N*	%	*N*	%
All	798	100.0	2318	100.0	2410	100.0	5526	100.0
**Age Groups**								
<12 years	404	50.6	1184	51.1	1359	56.4	2947	53.3
≥12 years	394	49.4	1134	48.9	1051	43.6	2579	46.7
**Sex**								
Boys	410	51.4	1147	49.5	1203	49.9	2760	49.9
Girls	388	48.6	1171	50.5	1207	50.1	2766	50.1
**SES**								
Low –Medium	511	64.0	1306	56.3	921	38.2	2738	49.5
High	287	36.0	1012	43.7	1489	61.8	2788	50.5
**Migrant Status**								
Both Parents	69	8.6	107	4.6	122	5.1	298	5.4
One parent	94	11.8	276	11.9	170	7.1	540	9.8
Neitherparent	635	79.6	1935	83.5	2118	87.9	4688	84.8
**Complementary Food Introduction**								
Missing	76	9.5	153	6.6	86	3.6	315	5.7
≤6 months	592	74.2	1892	81.6	1833	76.1	4317	78.1
>6 months	130	16.3	273	11.8	491	20.4	894	16.2
**Sweet Taste Preference**								
Low	350	43.9	950	41.0	918	38.1	2218	40.1
High	448	56.1	1368	59.0	1485	61.6	3301	59.7
Missing ^1^	0	0	0	0	7	0.3	7	0.1
**Fatty Taste Preference**								
Low	317	39.7	855	36.9	948	39.3	2120	38.4
High	481	60.3	1463	63.1	1460	60.6	3404	61.6
Missing ^1^	0	0	0	0	2	0.1	2	0.0
**Bitter Taste Preference**								
Low	546	68.4	1583	68.3	1624	67.4	3753	67.9
High	223	27.9	666	28.7	707	29.3	1596	28.9
Missing ^1^	29	3.6	69	3.0	79	3.3	177	3.2
**Country**								
Italy	170	21.3	652	28.1	224	9.3	1046	18.9
Estonia	57	7.1	265	11.4	554	23.0	876	15.9
Cyprus	228	28.6	508	21.9	169	7.0	905	16.4
Sweden	41	5.1	208	9.0	433	18.0	682	12.3
Germany	181	22.7	310	13.4	315	13.1	806	14.6
Hungary	60	7.5	242	10.4	509	21.1	811	14.7
Spain	61	7.6	133	5.7	206	8.5	400	7.2

¹ Missing values were generated when calculating the taste preference scores for each taste group. N is the number of participants in the BF&FMF categories.

**Table 3 nutrients-11-01040-t003:** The association between exclusive breastfeeding (EBF) duration and taste preferences in 6–16 year old children and adolescents who participated in the IDEFICS/I.Family studies.

	Sweet Taste (*N* = 5191)			Fatty Taste (*N* = 5196)			Bitter Taste (*N* = 5029)		
	OR	95% CI		OR	95% CI		OR	95% CI	
**EBF**									
(ref: None)	1.00			1.00			1.00		
≤4 months	1.12	0.97	1.29	1.04	0.90	1.21	0.98	0.84	1.15
>4 months	1.10	0.95	1.29	1.02	0.87	1.19	0.95	0.80	1.12
Healthy Diet Adherence Score (HDAS) ^$^	0.88	0.82	0.96	0.88	0.81	0.95	1.31	1.20	1.43
**Complementary Food Introduction**									
(ref. ≤6 months)									
>6 months	0.91	0.77	1.07	0.81	0.68	0.95	1.02	0.86	1.22
Missing	0.88	0.67	1.14	0.82	0.62	1.07	1.02	0.75	1.38
**SES**									
(ref. high)									
Low- medium	1.11	0.98	1.26	1.14	1.009	1.30	0.88	0.77	1.017

^$^ OR is calculated for difference in one unit from the mean. (Logistic regression models were adjusted also for age, sex, migrant background and country—OR not reported).

**Table 4 nutrients-11-01040-t004:** The association between combined breastfeeding (BF&FMF) duration and taste preference in 6–16 year old children and adolescents who participated in the IDEFICS/I.Family studies.

	Sweet Taste (*N* = 5191)			Fatty Taste (*N* = 5196)			Bitter Taste (*N* = 5029)		
	OR	95% CI		OR	95% CI		OR	95% CI	
**BF&FMF**									
(ref. None)	1.00			1.00			1.00		
≤6 months	1.11	0.93	1.32	1.15	0.96	1.37	0.98	0.81	1.19
>6 months	1.10	0.92	1.32	1.03	0.85	1.24	1.09	0.89	1.33
HDAS ^$^	0.88	0.81	0.96	0.88	0.81	0.95	1.31	1.20	1.43
**Complementary Food Introduction**									
(ref. ≤6 months)									
>6 months	0.91	0.78	1.07	0.82	0.70	0.97	0.99	0.83	1.18
Missing	0.88	0.67	1.15	0.82	0.63	1.08	1.01	0.74	1.37
**SES**									
(ref. High)									
Low- medium	1.11	0.98	1.26	1.14	1.004	1.29	0.90	0.78	1.03

^$^ OR is calculated for difference in one unit from the mean. (Logistic regression models were adjusted also for age, sex, migration status and country—OR not reported.

**Table 5 nutrients-11-01040-t005:** The association between exclusive formula milk feeding (EFMF) duration and taste preference in 6–16 years old children and adolescents who participated in the IDEFICS/I.Family studies.

	Sweet Taste (*N* = 5191)			Fatty Taste (*N* = 5196)			Bitter Taste (*N* = 5029)		
	OR	95% CI		OR	95% CI		OR	95% CI	
**EFMF**									
(ref. None)									
≤6 months	1.001	0.84	1.18	1.09	0.92	1.29	0.85	0.71	1.02
>6 months	1.009	0.85	1.18	1.008	0.85	1.18	0.95	0.80	1.13
HDAS ^$^	0.88	0.82	0.96	0.88	0.81	0.96	1.31	1.20	1.43
**Complementary Food Introduction**									
(ref. ≤6 months)									
>6 months	0.91	0.78	1.07	0.81	0.69	0.95	1.004	0.84	1.19
Missing	0.87	0.67	1.14	0.82	0.62	1.07	1.01	0.75	1.37
**SES**									
(ref. High)									
Low- medium	1.10	0.97	1.25	1.14	1.004	1.29	0.90	0.78	1.03

^$^ OR is calculated for difference in one unit from the mean. (Logistic regression models were adjusted also for age, sex, migration status and country—OR not reported).

**Table 6 nutrients-11-01040-t006:** The latent class profiles and highest conditional probabilities (ρ-estimate) of categories within each profile.

**Latent Profiles of Feeding Practices**			**Frequency**	**%**
**1. Long period of EBF and mixed breastfeeding, no exclusive use of formula milk, late introduction of complementary foods**			1334	24.14
Variable	Category	ρ-estimate		
Mixed feeding	more than 6 months	0.869		
Exclusive breastfeeding	more than 4 months	0.853		
Exclusive formula milk feeding	None	0.952		
Introduction of vegetables	after month 6	0.674		
Introduction of fruit	after month 6	0.562		
Introduction of meat	after month 6	0.981		
Introduction of cow milk	after month 6	0.988		
**2. Predominantly formula milk feeding (mixed and exclusive) and late introduction of complementary foods**			774	14.01
Variable	Category	ρ-estimate		
Mixed feeding	0–6 months	0.702		
Exclusive breastfeeding	0–4 months	0.464		
Exclusive formula milk feeding	more than 6 months	0.619		
Introduction of vegetables	after month 6	0.832		
Introduction of fruit	after month 6	0.657		
Introduction of meat	after month 6	0.997		
Introduction of cow milk	after month 6	0.973		
**3. Short duration of EBF and mixed BF without exclusive formula milk use, early introduction of main complementary foods**			2700	48.86
Variable	Category	ρ-estimate		
Mixed feeding	0–6 months	0.541		
Exclusive breastfeeding	0–4 months	0.627		
Exclusive formula milk feeding	None	0.764		
Introduction of vegetables	before month 6	0.964		
Introduction of fruit	before month 6	0.988		
Introduction of meat	before month 6	0.681		
Introduction of cow milk	after month 6	0.824		
**4. No breastfeeding, but exclusive formula milk use, early introduction of main complementary foods.**			718	12.99
Variable	Category	ρ-estimate		
Mixed feeding	None	0.841		
Exclusive breastfeeding	None	0.994		
Exclusive formula milk feeding	more than 6 months	0.583		
Introduction of vegetables	before month 6	0.911		
Introduction of fruit	before month 6	0.983		
Introduction of meat	before month 6	0.645		
Introduction of cow milk	after month 6	0.806		

**Table 7 nutrients-11-01040-t007:** The results of logistic regression models investigating the association between latent profiles of feeding practices (FP) and taste preferences of 6–16 year old children and adolescents who participated in the IDEFICS/I.Family studies.

	Sweet Taste (*N* = 5191)			Fatty Taste (*N* = 5196)			Bitter Taste (*N* = 5029)		
Variables	OR	95% CI		OR	95% CI		OR	95%CI	
Profile of FP									
(ref: profile 4)	1.00			1.00			1.00		
Profile 1	0.99	0.81	1.20	0.97	0.79	1.19	1.09	0.88	1.36
Profile 2	0.90	0.72	1.13	1.13	0.90	1.43	1.05	0.82	1.35
Profile 3	1.09	0.91	1.30	1.11	0.92	1.33	1.06	0.87	1.29
HDAS ^$^	0.88	0.82	0.96	0.88	0.81	0.95	1.31	1.20	1.43
SES (ref: high)									
Low/medium	1.11	0.97	1.25	1.14	1.004	1.29	0.89	0.78	1.02

^$^ OR is calculated for difference in one unit from the mean. (Logistic regression models were adjusted also for age, sex, migration status and country—OR not reported).

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
