# Peer review of "Association of Infant Feeding Patterns with Taste Preferences in European Children and Adolescents: A Retrospective Latent Profile Analysis"

_nutrients, 2019, doi:10.3390/nu11051040_

Reviewer 1 Report

Line 45. Provide reference for the first sentence.

Line 133. Why was a cut-off of 12 used to determine the person completing the questionnaire?

Line 140. Aside from the pre-testing to confirm the children were familiar with the foods, has the food and beverage questionnaire been validated?

Limitations are identified and communicated clearly but use of non-validated surveys should be added if any of the survey methods have not been tested for validity and reliability.

Author Response

Dear Reviewer,

Please find attached the point-by-point responses.

Reviewer 2 Report

Abstract

·         I would prefer if the abstract did not have so many abbreviations in it, as it is difficult to follow.

·         Conclusion of abstract: mention that prospective research is required.

Introduction

·         Paragraph 1: This is not completely clear. Desire for sweet tastes and rejection of bitter tastes are innate. Taste preference can be further developed/overcome through exposure and familiarity. It is worth naming and explaining all the basic tastes, especially as you have chosen to asses “fatty” taste preference, which is not traditionally viewed as one of the five basic tastes.

·         Lines 69-75: “Various types and brands of formula milk products – in particular in different EU countries - offer a vast diversity of flavors which formula-fed infants learn 70 to prefer depending on the formula milk they were fed with. This has been found to influence taste preferences later in life [19,20]. Infant formulas might differ in protein, fat and carbohydrate composition and/or structure and these differences may in turn affect growth, health outcomes and taste preferences [21].” : Are you referring to standard infant formula or specialist infant formula (e.g. hydrolysed formula for cows’ milk allergy?. I would disagree that standard infant formula has a vast diversity of flavours – do you have evidence for this statement? Nutritional composition in Europe is also strictly governed by legislation. Please clarify or modify this text.

·         Lines 88-91: “Although research has shown that breastfeeding influences infant’s food acceptance [16,33-35], to our knowledge, no other studies have examined if it shapes taste preferences in later stages of 90 life.”: Please clarify somewhere in the introduction that you are referring to normal healthy developing infants, as  studies have taken place examining taste preferences in those fed specialised diets/formulas for medical conditions.

Methods

·         Line 145: “For the present analyses, the 145 preferences for sweet and fatty taste were used as proxy-indicator of unhealthy foods and bitter taste as proxy-indicator of healthy foods.” Please reference this statement. What about salty foods? It was stated in previous paragraph that the questionnaire would assess preference for sweet, fatty, salty and bitter.

·         Also need to explain why you are not assessing umami or sour tastes.

·         Line 170: During the IDEFICS surveys (2007/2008 and 2009/2010) and the I.Family survey (2013/2014), mothers were asked to report on feeding strategies they chose during infancy and early stage of life of their children”: was this data collected prospectively or retrospectively ? It would be useful if the data collection processes were listed in chronological order, either in text or a flowchart.

·         Lines 240 and throughout the manuscript: “High fat” and “High sweet” – do not need capital H.

Discussion

·         Lines 336: “Thus, public health education programs should emphasize the role of food variety in shaping children’s preferences for bitter tasting foods in a long term and target mothers of low to medium socio-economic status in order to reduce their children’s preferences for high energy-dense foods”. Only mothers?! What about fathers and the extended family?

·         Retrospective feeding data: highly subject to recall bias. Please read and refer to literature on this.

·         Please discuss the role of the maternal diet (during breastfeeding) and the families’ diet as influential factors on your results. You have not measured either of these, so this should be mentioned in the limitations section.

·         What was the attrition rate/dropout rate at each stage of the study? I don’t see this mentioned in the results or the discussion. A flowchart showing number of participants recruited at each stage would be beneficial.

Author Response

Dear Reviewer,

Please find attached our point-by-point responses.

Nutrients EISSN 2072-6643 Published by MDPI AG, Basel, Switzerland RSS E-Mail Table of Contents Alert
Back to Top